# The Association of Different Genetic Variants with the Development of Hypoxic–Ischemic Encephalopathy

**DOI:** 10.3390/biomedicines11102795

**Published:** 2023-10-15

**Authors:** Vesna Pavlov, Anet Papazovska Cherepnalkovski, Marino Marcic, Ljiljana Marcic, Radenka Kuzmanic Samija

**Affiliations:** 1Department of Neonatology, Clinic for Gynecology and Obstetrics, Clinical Hospital Center Split, 21000 Split, Croatia; anet.cherepnalkovski@gmail.com; 2University Department of Health Studies, University of Split, 21000 Split, Croatia; 3Department of Neurology, Clinical Hospital Center Split, 21000 Split, Croatia; marino.marcic@yahoo.com; 4Department of Radiology, Polyclinic Medikol, 10000 Zagreb, Croatia; ljiljana.marcic@mefst.hr; 5Department of Pediatrics, Clinical Hospital Center Rijeka, 51000 Rijeka, Croatia; dadakuzmanicsamija@gmail.com

**Keywords:** hypoxic–ischemic encephalopathy, genetic variants, single nucleotide polymorphisms

## Abstract

The aim of this study is to investigate the frequency of six tag SNPs (single nucleotide polymorphisms) within specific genes (*F2*, *F5*, *F7*, *MTHFR*, *NOS2A*, *PAI 2-1*, *PAI 2-2*, and *PAI 3-3*)*: F2* (*rs1799963*), *F5* (*rs6025*), *F7* (*rs6046*), *NOS 2* (*rs1137933*), *PAI 2* (*SERPINB2*) (*rs6103*), *MTHFR* (*rs1801133*). The study also investigates their association with the development and severity of HIE. The genes *F2*, *F5*, and *F7* code for proteins involved in blood clotting. *MTHFR* is a gene that plays a significant role in processing amino acids, the fundamental building blocks of proteins. *NOS2A*, *PAI 2-1*, *PAI 2-2*, and *PAI 3-3* are genes involved in the regulation of various physiological processes, such as the relaxation of smooth muscle, regulation of central blood pressure, vasodilatation, and synaptic plasticity. Changes in these genes may be associated with brain injury. This retrospective study included 279 participants, of which 132 participants had Hypoxic–Ischemic Encephalopathy (HIE) and 147 subjects were in the control group. Our study found that certain genetic variants in the *rs61103* and *rs1137933* polymorphisms were associated with hypoxic–ischemic encephalopathy (HIE) and the findings of the magnetic resonance imaging. There was a correlation between Apgar scores and the degree of damage according to the ultrasound findings. These results highlight the complex relationship between genetic factors, clinical parameters, and the severity of HIE.

## 1. Introduction

Perinatal hypoxic–ischemic encephalopathy (HIE) is a serious brain injury that occurs in newborns and is characterized by a lack of oxygen (hypoxia) and insufficient blood flow (ischemia) to the brain, leading to irreversible neurological damage [1,2]. It is a type of brain damage that occurs when an infant’s brain does not receive enough oxygen and blood [2]. This condition is usually caused by a lack of oxygen during birth, which can happen due to a variety of reasons such as a prolonged labor, umbilical cord problems, or placental abnormalities [2]. Some genes are more interesting in this regard, such as six SNPs (single-nucleotide polymorphisms) in the genes *F2: rs1799963*, *F5: rs6025*, *F7: rs6046*, *NOS2: rs1137933*, and *PAI2: rs6103* [3]. The genes *F2*, *F5*, and *F7* code for proteins involved in blood clotting [4,5,6,7]; *MTHFR* is a gene that plays a significant role in processing amino acids [8], and *NOS2*, *PAI 2* are genes involved in the regulation of various physiological processes such as the relaxation of smooth muscle, regulation of central blood pressure, vasodilatation, and synaptic plasticity [9,10]. Their mutations lead to impaired functions of the enzymes that regulate the above-mentioned processes, and consequently to disorders in the regulation of the blood clotting system, blood pressure regulation, and intracranial hemorrhage and are associated with the development and severity of HIE [11].

The aim of this study was to determine the frequency of six SNPs within the genes *F2*, *F5*, *F7*, *MTHFR*, *NOS2*, and *PAI2* in relation to the occurrence and severity of perinatal hypoxic–ischemic brain injury, to determine the frequency of gene haplotypes in relation to the occurrence of perinatal hypoxic–ischemic brain injury and to determine the protective or risk-increasing role these genes play in the development of perinatal hypoxic–ischemic brain injury.

We chose the criteria for the selection of polymorphisms according to the topic of the study and based on the study of the current literature.

Genetic factors have long been considered a significant contributor to the development and progression of HIE, which is the focus of this study. However, what is particularly interesting is how certain genetic variants may be associated with its development. Sometimes, a single change in the genetic code can have a significant impact on human health.

By identifying specific genetic variants that are associated with an increased risk of hypoxic–ischemic encephalopathy, it may be possible to develop targeted interventions or therapies to prevent or mitigate the effects of this condition, identify new therapeutic targets, and improve prenatal care for at-risk fetuses.

Diagnosing HIE requires accurate observation and assessment via various techniques, including magnetic resonance imaging (MRI) and cranial ultrasonography [2]. MRI plays a crucial role in diagnosing and determining the severity of hypoxic–ischemic brain damage. Evaluation parameters such as the white matter’s altered signal, the size of the lateral ventricle anterior horns, the extension of subarachnoid space, the thinning of the corpus callosum, and the appearance of cystic changes are all observed and scored from 1 to 3 points, providing a tangible grade of brain injury [3,11].

HIE is a serious condition that requires immediate medical attention and requires monitoring of the child and numerous therapeutic interventions to prevent further brain damage [10].

## 2. Materials and Methods

Our study was conducted with approval of the Ethics Committee of University Hospital Split and School of Medicine, University of Split; class: 500-03/13-01/13. number: 2181-147-01/06/J.B.-13-1. We obtained informed consent from all patients’ parents prior to blood sampling.

The retrospective study included 279 participants. Out of the total number, 132 (47%) were participants affected with Hypoxic–Ischemic Encephalopathy (HIE), while 147 (53%) constituted the control group.

We conducted research on 279 unrelated children, 83 boys and 49 girls, who suffered from Hypoxic–Ischemic Encephalopathy (HIE). They were treated at the University Hospital Split, Children’s Hospital in the period of 1992–2008, and at the Department for Gynecology and obstetrics, Department for neonatology in the period of 2013–2015. Each child met at least one criterion such as abnormal fetal heart rate, low Apgar score (1–5), an arterial pH of 7.1 or lower, or evidence of hypoxic–ischemic brain damage found on cranial ultrasonography. HIE was evaluated using brain ultrasound. Brain ultrasound was performed in premature infants with a gestational age of less than 37 weeks on the first day of life and repeated at least once during the first week and at 2, 4, and 6 weeks of age. Brain ultrasound was performed in newborns born at 37 weeks of gestational age or later depending on clinical indications. Hypoxic–ischemic brain injury and intracranial hemorrhage were confirmed via magnetic resonance imaging at the age of 2 years.

A total of 3 mL of EDTA anticoagulated blood was sampled for the genetic analyses at the same time that routine blood samples were obtained during hospitalization. Genetic analyses were performed at the Medical Biology Laboratory at the University of Split School of Medicine.

Genomic DNA was extracted from peripheral blood leukocytes using the Perfect gDNA kit (Eppendorf, Hamburg, Germany). Six haplotype-tagging SNPs (*rs1799963*, *rs6025*, *rs6046*, *rs1137933*, *rs6098*, *rs6103*, *rs6104*) were examined. Real-time polymerase chain reaction (PCR) analysis for the polymorphisms was performed using an ABIPRISM 7500 Sequence Detection System (Applied Biosystems, Foster City, CA, USA) and pre-developed TaqMan assay reagents. PCR was carried out according to the manufacturer’s protocol. Genomic DNA was isolated from peripheral blood leucocytes using QIAamp DNA Blood Mini Kit-ova (Qiagen, Hilden, Germany). Peripheral blood was taken from all children. Blood samples (5–7 mL) were taken in VacutainerR test tubes (Becton Dickinson, Meylan, France) with EDTA anticoagulant, and until isolation of DNA, they were kept at −20 °C. The isolation method consisted of several stages: breaking cells and nuclei, destruction of all cellular proteins, ribonucleic acid, and other macromolecules, releasing and cleansing of DNA of the remaining proteins, and DNA storage. The concentration of isolated DNA was measured on a Nanodrop spectrophotometer (Thermo Scientific, Langensel Bold, Germany). The average concentration was around 30 μg/mL. For the genotyping method, the sample was diluted to a concentration of 10 ng/μL.

For full-term infants with brain damage, the diagnosis was confirmed via regular cranial ultrasonography conducted at 3, 7, 14, and 21 days after delivery, and then monthly until the anterior fontanel closed. These assessments were performed by independent experts, neonatologists with more than 10 years of experience.

To asses and follow the development of HIE in all cases, we used an ultrasound device, Siemens, Healthineers, Acuson NX3 Elite with a convex transducer frequency of 7.3 MHz.

The ultrasound findings included a description of the ventricular system, midline structures, parenchimal echogenicity, and posterior fossa structures. Ventricular dilatation was estimated according to measurement of the anterior horn with and of the thalamo –occipital distance. The ultrasound findings were classified as follows: category 1, normal and mild abnormalities: asymmetric lateral ventricles, mild dilatation of the occipital horns (thalamo–occipital distance ˂ 95. percentile), cysts on the choroid plexus, frontal, temporal and caudothalamic pseudocysts, and lenticulostriate vasculopathy; category 2, severe abnormalities: PVL type 2, HIC type 2; and category 3, very severe abnormalities: PVL type 3, HIC type 3 and 4, GHM-IVH. Definitions were in accordance with the Papiles criteria (1), venous /arterial stroke, and malformations.

At age 2, we determined the final diagnosis and grade of brain damage using magnetic resonance imaging (MRI). MR is performed by a neuroradiologist with more than 10 years of experience.

The diagnosis of the final neuroanatomical outcome and degree of hypoxic–ischemic damage in the examined group was determined by means of MRI of the brain at the age of two years, when the processes of proliferation, migration, organization, and myelination were completed. The children were imaged using an MR device: Siemens AG, Erlangen, Germany Magnetom Symphony, 1.5 T. Changes in the white matter signal, appearance of cystic changes, changes in the size of the anterior horns of the lateral ventricles, enlargement of the subarachnoid spaces, and thinning of the corpus callosum were monitored (2). Damage scoring was adjusted according to previously published research (2). Each parameter that was observed was scored from 1–3 points, so that subjects with up to 5 points have a good result or grade 1, grade 2 or medium-severe damage was scored from 6 to 10 points, and grade 3 or very severe damage from 11 to 15 points.

Data on demographic characteristics such as gender, gestational age (gestational age 37 and over; 33 to 36 weeks; and less than 32 weeks), and birth weight were collected from medical records for both investigated groups. Patients with congenital malformations of the central nervous system, metabolic diseases, chromosomal abnormalities, central nervous system infections, and trauma patients were excluded from the study. Also, newborns from multiple gestations, and newborns of mothers who had pre-eclampsia and placental abruption were excluded.

We observed changes such as altered white matter signal, appearance of cystic changes, size alterations in the lateral ventricle anterior horns, extension of subarachnoid space, and thinning of the corpus callosum. We collected demographic data and other characteristics such as gender, gestational age, and birth weight from the medical records. The participants were divided into subgroups for analysis based on gestational age, birth weight, and Apgar score. We excluded patients with congenital malformations, metabolic diseases, chromosomal abnormalities, infections, CNS trauma, multiple gestations, and infants of mothers who had pre-eclampsia and placental abruption.

Out of the HIE group, 132 patients (72.7%) had hypoxic–ischemic injury and 30 patients (27.3%) had intracranial hemorrhage. We also had a control group of 147 healthy children—76 boys and 71 girls—with normal brain ultrasound findings and/or normal neurological status after the second year of life.

Genotype frequencies were examined, Hardy–Weinberg equilibrium (HWE) was maintained, and minor allele frequencies (MAF) were scrutinized.

We analyzed the sensitivity of the genotype and conducted a case–control and a cis-position haplotype analysis of 6 SNPs using Haploview 4.1.

We also analyzed marker single nucleotide polymorphisms (SNPs) within the genes: *F2: rs1799963*, *F5: rs6025*, *F7: rs6046*, *Nitric oxide synthase 2a isoform1: rs1137933*, *PAI2-1: rs6098*, *PAI2-2: rs6103*, and *PAI2-3: rs6104*. Genotyping was performed using polymerase chain reaction (PCR) in real time. Association analyses were performed according to allelic and genotypic distribution.

### Statistical Analysis

Chi-square tests were used to study the genotypic and allelic distribution for each polymorphism, considering factors like gestational age, gender, Apgar score, birth weight, and MRI findings.

We used Statistica 7.0 (http://statistica.software.informer.com/7.0/ (accessed on 1 March 2023)) for logistic regression to calculate ORs, helping estimate the likelihood of hypoxic–ischemic brain damage based on different factors. A multiple logistic regression was performed focusing on haplotype distribution. Chi-square tests were used again to compare the extent of brain damage.

## 3. Results

According to the Hardy–Wienberg equilibrium, genotype frequencies in a population remain constant between generations in the absence of disturbance by outside factors. All genes and their polymorphisms that we examined in our study are compatible with the Hardy–Weinberg principle, as shown in Table 1.

Table 1 shows the genotypic analysis of polymorphisms of the tested SNPs in Hardy–Weinberg equilibrium (HWE).

Table 2 shows a comparison of the frequency of the less common alleles between the diseased and control groups for the investigated polymorphisms.

The results of the association analysis of the less common alleles between the diseased and control group for the investigated polymorphisms. The difference in the frequency of rare alleles between the affected and control groups was analyzed for the six studied SNPs. The difference in frequency between the affected and control groups for SNP *rs6103*, allele G, was determined. The results are shown in Table 2. No difference was observed for the other SNPs.

The retrospective study included 279 participants. Out of the total 279 participants, 132 (47%) were HIE patients, while 147 (53%) comprised the control group. Table 3 presents the demographic data of the study groups.

The distribution of children according to gender did not show a statistically significant difference between the investigated groups (*p* = 0.078). The median birth weight of children in the control group was higher by 205 g compared with the affected group (*p* = 0.005). The median gestational age was longer by one month in the control group compared with the affected group (*p* < 0.001). The median Apgar score at 1 min was higher in the control group compared with the affected group (*p* < 0.001). Table 4 shows the distribution of genotypes for the six investigated polymorphisms among the studied groups.

The distribution of alleles significantly differed between the affected and control groups in the *rs61103*, *rs6103* polymorphism (chi-square = 7.3; *p* = 0.026). The frequency of the GG allele was three times higher in the control group compared with the affected group. The odds of GG occurrence compared with CC occurrence were 3.8 times higher in the control group than in the affected group (OR: 3.8; 95% CI: 1.3–11; *p* = 0.013).

There was no statistically significant difference in the distribution of alleles between the affected and control groups in the other investigated polymorphisms (*p* > 0.05).

Table 5 shows the distribution of demographic parameters in the affected group in relation to the ultrasound (US) findings and MRI gradings.

The distribution of demographic parameters in the affected group did not show a statistically significant difference in relation to the ultrasound (US) findings.

In the group of patients, only two children had a normal finding. Only alleles CC and CG were included in the logistic regression due to a small number of GG alleles. There is a statistically significant association between Apgar score groups and the degree of damage according to ultrasound findings. The proportion of Apgar scores ≤ 5 is 2.2 times higher in the grade 3 group compared with the grade 2 group (χ^2^ = 8.7; *p* = 0.034). The odds of grade 3 damage occurring compared with grade 2 damage according to ultrasound findings increase by 1.6 times with each decrease in the Apgar score category (OR: 1.6; 95% CI: 1–2.5; *p* = 0.039).

A statistically significant association of gestational age (GA), gender (G), birth weight (BW), and alleles with the degree of damage was not demonstrated (*p* > 0.05).

Table 6 shows the distribution of genotypes of polymorphism rs6103 according to clinical parameters, gender, and ultrasound findings.

The analysis of all variables presented in Table 4 was performed for the CC and CG genotypes due to the small sample size in the GG allele. There was no statistically significant association found between gestational age, gender, Apgar score, and birth weight with the CC and CG alleles (*p* > 0.05) in the rs6103 polymorphism.

Table 7 shows the distribution of demographic parameters among the affected individuals in relation to MRI findings.

Table 8 shows the association between the rs1137933 polymorphism and magnetic resonance imaging (MRI).

There is a statistically significant association between the rs1137933 polymorphism and magnetic resonance imaging (MRI) findings. None of the AA alleles were found in grade 3 MRI results. The frequency of AG alleles is twice as high in grade 3 MRI images compared with grade 2 MRI scans. Since there are only two AA alleles, we compared the AG and GG alleles and found a difference between them in relation to the degree of damage according to MRI at a significance level of 90% (chi-square = 2.7; *p* = 0.102). Table 9 shows the association between any of the polymorphisms and the US findings.

There was no statistically significant association found between any of the polymorphisms and US findings (*p* > 0.05).

## 4. Discussion

Our study demonstrated an association between the SNP rs1137933 and HIE. The proportion of AG alleles was twice as high in grade 3 on the magnetic resonance images compared with grade 2. The SNP rs1808593 showed an association with genotype, and the rs1800783-rs1800779 TG haplotype showed an association with HIE. The distribution of alleles differed significantly between the affected and control groups in the rs61103 polymorphism. The proportion of GG alleles was three times higher in the control group than in the affected group.

Several recently published articles have applied a similar approach to the analysis of tagging polymorphisms but in correlation with other vascular disorders [12,13,14].

In the Genetics-of-Early-Onset Stroke (GEOS), a case–control study of ischemic stroke in a biracial population among men and women aged 15 to 49 years (Cole et al.), the association of PROCR rs9574 polymorphisms with early-onset ischemic stroke was established in Caucasians. PROCR, which was in strong LD (interconnected, near each other) with eight other SNPs and one independent SNP rs2069951, was significantly associated with ischemic stroke [13].

Esih et al. investigated IL1B rs16944 allele polymorphisms and the development of epilepsy in children with HIE and found that there is an increased risk of brain damage and the development of cerebral palsy in CARD8-IL1B gene–gene interaction. They also observed that the IL1B rs16944 promoter polymorphism can cause a stronger neurological response and consequently a worse neurological outcome after hypoxic ischemic encephalopathy treated with therapeutic hypothermia [15]. Unlike them, we proved a significant association between the rs1137933 polymorphism and magnetic resonance imaging findings. But AA alleles were found in grade 3 on the magnetic resonance imaging, which could indicate a protective role in terms of brain damage. The proportion of AG alleles was twice as high in grade 3 compared with grade 2 on the MRI scans. Since there were only two AA alleles, AG and GG alleles were compared, and a difference in relation to the degree of damage according to the MRI finding was obtained.

We proved a significant association between Apgar score groups and the degree of damage according to ultrasound findings. The proportion of Apgar scores ≤ 5 was 2.2 times higher in the grade 3 group than in the grade 2 group. The odds ratio for the occurrence of grade 3 compared with grade 2 of damage according to ultrasound findings increased by 1.6 times with each decrease in the Apgar score category. Previous studies have shown a higher prevalence of cerebral palsy in children with Apgar scores < 3, showing the prevalence to be 130 times higher than in children with Apgar scores of 10 [16].

Lie et al. studied the association of the Apgar score 5 min after birth with cerebral palsy in children of normal body weight and children with low birth weight, as well as the association with diagnoses of cerebral palsy. They found that the Apgar score was strongly associated with cerebral palsy. This association was high in children with normal birth weight and modest in children with low birth weight [16].

We established an association between the rs1137933 polymorphism and magnetic resonance findings. The proportion of AG alleles was twice as high in grade 3 compared with grade 2 on the magnetic resonance. Since there were only two AA alleles, AG and GG alleles were compared, and a difference in relation to the degree of damage according to the magnetic resonance was obtained.

Guarnera et al. studied MRI findings in children with HIE and emphasized the importance of the DWI/ADC sequence for the evaluation of brain damage in children. ADC values can be used as prognostic biomarkers to predict neurodevelopmental outcomes in children and provide parents with the most necessary information related to patient prognosis [17].

Numerous studies have investigated the association of polymorphisms in the NOS3 gene with the occurrence of cerebral palsy, but only two studies have confirmed it [18,19,20].

The capacity for greater plasticity in the developing brain is a major difference between the nervous system in infants and children compared with adults. Some common pediatric disorders are disrupted by the molecular signaling pathways involved in brain plasticity. Better treatments for currently untreatable disorders may be achieved due to these discoveries. The overstimulation of these plasticity mechanisms usually occurs as a reaction to brain damage in hypoxia–ischemia. A better understanding of the molecular pathways involved in plasticity and injury can help us in developing pediatric neurology [21].

Gibson et al. examined whether selected genetic polymorphisms in infants were associated with spontaneous preterm birth (<37 weeks) in children with or without later-diagnosed cerebral palsy. They found that the beta-adrenergic receptor and nitric oxide synthase variants are associated with prematurity and that genetic variants of the placental antifibrinolytic inhibitor of plasminogen activator 2, as well as thrombomodulin and alpha aducin, may contribute to the risk of spontaneous preterm birth [14].

Kuzmanic Samija et al. proved the genotypic and haplotypic associations of NOS3 polymorphisms with HIE [19].

Cole et al. investigated the association of eNOS gene promoter polymorphism T-786C with the cause of respiratory distress and IVH in premature infants. Blood samples from 124 African-American preterm infants were studied for the association between the mutated allele-786C and certain conditions affecting premature infants. The mutated allele-786C was present in 15.3% of preterm infants with respiratory distress syndrome, bronchopulmonary dysplasia, and IVH, compared with 7.25% of preterm infants without these conditions. A significant two-fold increase in the mutant allele in patients compared with controls (*p* = 0.04, OR 2.3) reveals that the eNOS-786C allele could be a significant risk factor in the development of respiratory conditions and IVH in premature infants [13]. In logistic regression, considering these SNPs simultaneously in non-Hispanic individuals, an association with CP was observed for eNOS-922 heterozygotes (OR 3.0, CI 1.4–6.4), F7 (OR 2.7, CI 1.1–6.5), LTA (OR 2.1, CI 1.0–4.6), and PAI-1 (OR 3.2, CI 1.2–8.7) [20].

In recent years, various neuroprotective strategies have been described using numerous in vitro and animal models, as well as clinical trials. Several pharmacological compounds have been proposed for perinatal asphyxia (PA) therapy. Amongst these drugs, NOS inhibitors may play an important role, since the destructive mechanisms of NO drivers are involved in all phases of PA. Clinical trials for 2-IB (2-iminobiotin) are now underway and will reveal whether NOS inhibition can be used as a therapeutic treatment for perinatal asphyxia [22,23,24,25,26,27,28].

The results of the study by Esih et al. showed that the severity of brain injury in newborns with HIE treated with therapeutic hypothermia significantly depends on the interaction between genes that are important in inflammatory and antioxidant pathways [15]. In our work, we proved that the genetic polymorphism of certain genes significantly affects the outcome of HIE measured by the damage detected via ultrasound and MRI, and Esih et al. found a link between genetic polymorphism and the development of epilepsy in these children.

The weaknesses of our study are that a limited number of genes were examined, that no genotyping and analysis of an independent cohort of different origins was performed, and that the study was not multiethnic. Therefore, there is a further need for additional research that would include more different genes, more polymorphisms of the examined genes, and populations from different parts of the world.

## 5. Conclusions

Our study showed how genetic polymorphisms for certain genes, i.e., *F2*, *F5*, *F7*, *MTHFR*, *NOS2A*, *PAI2-1*, *PAI 2-2*, and *PAI3-3*, can be significant in hypoxic brain injury both clinically, diagnostically, and therapeutically. Early and multidisciplinary perinatal assessment and subsequent reassessment of children are essential in order to identify physical and neuropsychological disorders to guarantee early and tailored therapy. Further studies are needed to address these key topics and confirm our results.

## Figures and Tables

**Table 1 biomedicines-11-02795-t001:** Genotypic analysis of polymorphisms of the tested SNPs in Hardy–Weinberg equilibrium (HWE).

SNP *	χ^2^	*p*-Value
rs6046	1.42	0.23
rs6025	2.76	0.10
rs6103	0.24	0.62
rs1137933	0.15	0.70
rs1799963	4.36	0.04
rs1801133	0.39	0.53

* SNP—single nucleotide polymorphisms, χ^2^—chi-square test, *p*-value. The genotypes of all six tested SNPs were in HW balance.

**Table 2 biomedicines-11-02795-t002:** Comparison of the frequency of the less common alleles between the diseased and control groups for the investigated polymorphisms.

SNP *	Rare Allele	MAF ** (HIE Group)	MAF (Control Group)	OR (95% CI)	*p*-Value
rs6046	A	0.125	0.1463	0.837 (0.517, 1.356)	0.471
rs6025	T	0.0227	0.0204	1.119 (0.352, 3.558)	0.849
rs6103	G	0.2197	0.3163	0.613 (0.419, 0.899)	0.012
rs1137933	A	0.1970	0.2483	0.722 (0.473, 1.102)	0.131
rs1799963	A	0.0189	0.0340	0.539 (0.179, 1.621)	0.271
rs1801133	A	0.3788	0.3605	1.099 (0.769, 1.571)	0.603

* SNP—single nucleotide polymorphisms, ** MAF—minor allele frequency; χ^2^—chi-square test; *p*-value.

**Table 3 biomedicines-11-02795-t003:** Demographic data in relation to the study groups.

		HIE Group	Control Group	
G (%)				χ^2^ = 3.1; *p* = 0.078
	M	83 (63)	76 (52%)	
	F	49 (37)	71 (48)	
BW *	Median (Q1–Q3; min-max)	1695 (1200–2150; 580– 4390)	1900 (1340–3340; 400–4470)	Z = 2.8; *p* = 0.005
GA **	Median (Q1–Q3; min-max)	32 (28–35); 25–41	33 (30–40) (24–41)	Z = 3.5; *p* < 0.001
AS *****	Median (Q1–Q3; min- max)	6 (4–7); 0–10	7 (5–10; 2–10)	Z = 5.3; *p* < 0.001

* BW—birth weight in grams, ** GA—gestational age in weeks, ***** AS—Apgar score,); χ^2^—chi-square test and Mann–Whitney test.

**Table 4 biomedicines-11-02795-t004:** The distribution of genotypes for the six investigated polymorphisms among the studied groups.

SNP *	Genotype	HIE Group No (%)	Control Group No (%)	χ^2^; *p*-Value	OR (95% CI)	*p*-Value
rs6046						0.233
	AA *	4 (3%)	2 (1.4%)	2.97; 0.226		
	AG	25 (18.9%)	39 (26.5%)		0.321 (0.05–1.88)	0.208
	GG	103 (78%)	106 (72.1%)		0.486 (0.087–2.71)	0.410
rs6025	CC *	126 (95.5%)	141 (95.9%)	0.036; 0.849		
	CT	6 (4.5%)	6 (4.1%)		1.119 (0.352–3.558)	0.849
rs6103						0.034
	CC *	79 (59.8%)	71 (48.3%)	7.318; 0.026		
	CG	48 (36.4%)	59 (40.1%)		0.731 (0.44–1.2)	0.218
	GG	5 (3.8%)	17 (11.6%)		0.26 (0.09–0.753)	0.013
rs1137933						0.295
	AA *	4 (3%)	6 (4.1%)	2.45;		
	AG	44 (33.3%)	61 (41.5%)	0.294	1.082 (0.288–4.063)	0.907
	GG	84 (63.6%)	80 (54.4%)		1.575 (0.429–5.789)	0.494
rs1799963				1.24; 0.265		0.271
	AG *	5 (3.8%)	10 (6.8%)			
	GG	127 (96.2%)	137 (93.2%)		0.539 (0.179–1.621)	
rs1801133						0.863
	AA *	21 (15.9%)	20 (13.6%)	0.296; 0.862		
	GA	58 (43.9%)	66 (44.9%)		0.837 (0.413–1.697)	0.622
	GG	53 (40.2%)	61 (41.5%)		0.827 (0.405–1.691)	0.603

Note: * reference level. Data are reported as number with percent in parentheses, HIE: hypoxic–ischemic encephalopathy; OR: odds ratio; CI: confidence interval. χ^2^: chi-square test and logistic regression.

**Table 5 biomedicines-11-02795-t005:** The distribution of demographic parameters in the affected group in relation to the ultrasound (US) findings and MRI gradings.

Variables		MRI Grade According to US		Χ^2^; *p*-Value	OR (95%CI)	*p*-Value **
		2	3			
GA				1.24; 0.538	1 (0.59–1.7)	0.858
	≤32 *	19 (56)	54 (60)			
	33–36	11(32)	21 (23)			
	>37	4 (12)	15 (17)			
G				0.002; 0.163	1 (0.691–1.453)	0.992
	M *	21 (62)	56 (52)			
	F	13 (38)	34 (38)			
AS				8.7; 0.034	1.6 (1–2.5)	0.039
	≤5	8 (24)	47 (52)			
	6–7	18 (55)	28 (31)			
	8–9	4 (12)	11 (12)			
	10 *	3 (9)	4 (5)			
BW				2.4; 0.489	0.849 (0.57–1.3)	0.430
	≤1000 *	3 (9)	18 (20)			
	1001–1499	10 (29)	20 (22)			
	1500–2499	14 (41)	34 (38)			
	≥2500	7 (21)	18 (20)			
Allele rs6103				0.163; 0.922	0.494 (0.37–1.94)	0.599
	CC *	21 (62)	52 (58)			
	CG	12 (35)	35 (39)			
	GG	1 (3)	3 (3)			

** logistical regression, χ^2^—chi-square test and logistic regression were used for analyses; GA—gestational age, G—gender, AS—Apgar score, BW—body weight, OR—odds ratio, CI—confidence interval; MRI—magnetic resonance imaging, US—ultrasound, *—reference value.

**Table 6 biomedicines-11-02795-t006:** Distribution of genotypes of polymorphism rs6103 according to clinical parameters, gender, and ultrasound findings.

Variables		CC	CG	GG	Χ^2^; *p*-Value
GA (in weeks)					0.499; 0.779
	≤32	45 (60.8)	28 (58.3)	1 (25)	
	33–36	20 (27)	12 (25)	0	
	>37	9 (12.2)	8 (16.7)	3 (75)	
G					0.078; 0.780
	M	51 (65)	29 (60)	3 (60)	
	F	28 (35)	19 (40)	2 (40)	
AS					5.8; 0.120
	≤5	32 (43)	24 (51)	1 (25)	
	6–7	32 (43)	13 (28)	1 (25)	
	8–9	9 (12)	6 (13)	0	
	10	1 (2)	4 (8)	2 (50)	
BW (in grams)					2.1; 0.548
	≤1000	13 (18)	8 (16.7)	0	
	1001–1499	22 (30)	9 (18.8)	0	
	1500–2499	26 (34)	21 (43.8)	1 (25)	
	≥2500	13 (18)	10 (20.8)	3 (75)	

χ^2^—chi-square test, GA—gestational age, G—gender, AS—Apgar score, BW—body weight.

**Table 7 biomedicines-11-02795-t007:** Distribution of demographic parameters among the affected individuals in relation to MRI findings.

Variables		MRI Grade		OR (95%CI)	*p*-Value
		2	3		
	≤32 *	18 (60)	34 (64.2)		
	33–36	8 (26.7)	11 (20.8)		
	>37	4 (13.3)	8 (15.1)		
G				0.213	0.644
	M *	16 (53)	3 (61)		
	F	14 (42)	21 (39)		
AS				2.0	0.564
	≤5	14 (46.7)	27 (50.9)		
	6–7	8 (26.7)	18 (34)		
	8–9	5 (116.7)	6 (11.3)		
	10 *	3 (10)	2 (3.8)		
BW				0.90	0.825
	≤1000 *	4 (13.3)	8 (15.1)		
	1001–1499	6 (20)	15 (28.3)		
	1500–2499	13 (43.3)	19 (35.8)		
	≥2500	7 (23.3)	11 (20.8)		
Allele rs6103				0.8	0.400
	CC *	21 (70)	32 (59)		
	CG	7 (23)	20 (37)		
	GG	2 (7)	2 (4)		
US					
	2	9 (30)	14 (27)		
	3	21 (70)	38 (73)		

χ^2^: chi-square test and logistical regression, G: gender, AS: Apgar score, BW: body weight, OR: odds ratio, CI: confidence interval, MRI: magnetic resonance imaging, US: ultrasound, *: reference value.

**Table 8 biomedicines-11-02795-t008:** Association between the rs1137933 polymorphism and magnetic resonance imaging (MRI).

SNP	Genotype	MRI Grade 2	MRI Grade 3	OR (95% CI)	*p*-Value
rs6046					
	AA *	0	1 (1.9)	0.638	
	AG	7 (23.3)	11 (20.4)		0.727
	GG	23 (76.7)	42 (77.8)		
rs6025					
	CC *	27 (90)	53 (98%)		
	CT	3 (10)	1 (2%)		

rs6103					
	CC *	21 (70)	32 (59.3)	1.8	
	CG	7 (23.3)	20 (37)		0.400
	GG	2 (6.7)	2 (3.7)		
rs1137933					
	AA *	2 (6.7)	0	7.1	
	AG	7 (23.3)	25 (46.3)		0.028
	GG	21 (70)	29 (53.7)		
rs1799963					

	AG *	0	1 (2)		
	GG	30 (100)	53 (98)		
rs1801133	AA *	5 (16.7)	8 (14.8)	0.13,	
	GA	15 (50)	26 (48.1)		0.937
	GG	10 (33.3)	20 (37)		

χ^2^—chi-square test and logistical regression were used in analyses, OR—odds ratio, CI—confidence interval, MRI—magnetic resonance imaging, *—reference value.

**Table 9 biomedicines-11-02795-t009:** Association between any of the polymorphisms and US findings.

SNP	Genotype	US:2	US:3	χ^2^; *p*-Value	OR (95%CI)	*p*-Value
rs6046						
	AA *	1 (2.9)	3 (3.3)	0.106		
	AG	6 (17.6)	18 (20)			0.948
	GG	27 (79.4)	69 (76.7)			
rs6025						
	CC *	33 (97)	85 (94)			
	CT	1 (3)	5 (6)			

rs6103						
	CC *	21 (61.8)	0(57.8)	0.163		
	CG	12 (35.3)	35 (38.9)			0.922
	GG	1 (2.9)	3 (3.3)			
rs1137933						
	AA *	1 (2.9)	3 (3.3)	0.049		
	AG	12 (35.3)	30 (33.3)			0.976
	GG	21 (61.8)	57 (63.3)			
rs1799963						

	AG *	1 (3)	3 (3)			
	GG	33 (97)	87 (97)			
rs1801133						
	AA *	8 (23.5)	12 (13.3)	2.9		
	GA	16 (47.1)	39 (43.3)			0.237
	GG	10 (29.4)	39 (43.3)			

χ^2^— chi-square test and logistical regression, OR—odds ratio, CI—confidence interval, US—ultrasound, *—reference value.

## Data Availability

The authors of this study used publicly available electronic libraries at the University of Split.

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
