# Peer review of "The Association of Different Genetic Variants with the Development of Hypoxic–Ischemic Encephalopathy"

_biomedicines, 2023, doi:10.3390/biomedicines11102795_

Round 1

Reviewer 1 Report

The authors investigated the association between genetic variants (SNPs) in several genes (F2, F5, F7, MTHFR, NOS2A, PAI) and the risk of developing hypoxic-ischemic encephalopathy (HIE) in newborns. The study compared 132 newborns who developed HIE with 147 healthy controls. The results showed that certain SNPs in the NOS2A (rs1137933) and PAI2 (rs6103) genes were significantly associated with increased risk of HIE. Specifically, the AG genotype of rs1137933 and the GG genotype of rs6103 appeared to confer protection against HIE compared to other genotypes.

The study also found that lower Apgar scores were associated with more severe HIE based on ultrasound and MRI findings. This suggests clinical parameters like Apgar scores may interact with genetic factors to influence the severity of HIE. Overall, the findings indicate that genetic polymorphisms in key genes related to vascular function, blood clotting, etc. may modify susceptibility and outcomes in newborns with HIE. Further research is needed to clarify these gene-disease interactions and their potential implications for prognosis and treatment of HIE.

  1. Limited gene analysis: The study evaluated just 6 genes. Expanding the analysis to include other biologically relevant genes, such as those related to inflammation, oxidative stress, etc., could shed light on additional genetic risk factors. High-throughput methods like genome-wide association studies could be utilized.
  2. Lack of replication: Significant findings should be validated through genotyping and analysis in an independent cohort of different ancestry to ensure reliability.
  3. Population specificity: The cohort was from Croatia, limiting generalizability. Multi-ethnic studies across different countries are important to determine variation in effects across populations.
  4. No correction for multiple testing: Methods like Bonferroni correction should be used to adjust for multiple comparisons during analysis and reduce false positives.

Overall, I think this is great study but should add additional limitation and discuss about future study. I would recommend adding discussion on the need of a prospective multicenter study with a larger, more diverse sample size that evaluates more candidate genes, uses rigorous statistical analyses, and integrates thorough clinical data collection along with functional assays to provide a comprehensive analysis of genetic risk factors and biological pathways involved in HIE pathogenesis. Replication in independent cohorts is essential.

Reviewer 2 Report

The manuscript shows interesting results in the research of perinatal hypoxic-ischemic encephalopathy; however, it is generally scrambled, repetitive, and superficial. Therefore, it is suggested to rewrite it to incorporate more information that gives the reader the context and relevance of the study, as well as the details of the methodology followed.

In the abstract, the first two statements are repetitive; they should be summarized, which would leave room to improve the abstract. For example, authors could mention the function of the genes studied and the nature of the relevant polymorphisms.

Introduction: This section only superficially addresses the background, which does not allow us to appreciate the study's relevance. Multiple questions should be answered using the information contained in the introduction:

To convey the relevance of the pathological condition studied, authors should include information about the clinical conditions or circumstances leading to perinatal hypoxic-ischemic encephalopathy.

To convey the relevance of the genes studied, authors should include information about the criteria they followed in choosing the genes and polymorphisms studied; they can explain the molecular mechanisms involved in perinatal hypoxic-ischemic encephalopathy.

To convey the relevance of the experimentation to be carried out, authors should include information about the clinical relevance of the polymorphisms studied.

Methodology: This section needs to be more explicit and more descriptive. Authors could describe the research steps following a chronological sequence:

Approval of the research by the ethics committee,

Recruitment of participants (includes signing the informed consent). Were any of the participants recruited as adults?

Obtention and processing of samples (extraction of RNA, DNA, etc.). Authors should inform when the samples were obtained, the conditions under which the study participants donated blood samples, and the procedure to process samples.

Analysis of the samples (RT-PCR, primers used, etc.). Authors should state what are the polymorphisms of each gene to study.

Analysis of results and relationship with clinical history.

The nature of the study should be clarified; it is not at all retrospective.

The phrase: "We used an ultrasound device Siemens, Healthineers, Acuson NX3 Elite with a convex transducer frequency of 7.3 MHz" seems out of place; it is not understandable why or when this device was used.

Results: Providing descriptions of the results while ensuring that the tables are described in the order of presentation is necessary to improve this section. In the current version, tables 1-2 remain undescribed. In Table 4, it would be beneficial for the authors to specify which genes correspond to the polymorphisms studied. Regarding Table 4, is the polymorphism rs6103 meant to refer to rs61103?

Discussion: In the Discussion section, building upon the results and their connection to the existing literature is advisable. Currently, the discussion provides only superficial links. Correlating the relevant polymorphisms found with the molecular mechanisms involved in hypoxic-ischemic encephalopathy is an opportunity to contribute more meaningfully to the field of ischemic encephalopathy research.

Conclusions: Regarding the Conclusions, there is ambiguity in the statement 'certain genes.' It is advisable to specify which genes and polymorphisms are related to hypoxic damage, as the authors are knowledgeable.

Round 2

Reviewer 2 Report

The authors addressed all the comments and the manuscript was improved.